# Mammographic density mediates the protective effect of early-life body size on breast cancer risk

Marina Vabistsevits [1,2] ✉, George Davey Smith [1,2], Tom G. Richardson [1,2], Rebecca C. Richmond [1,2], Weiva Sieh [3,4], Joseph H. Rothstein[3,4], Laurel A. Habel[5], Stacey E. Alexeeff[5], Bethan Lloyd-Lewis [6,7] & Eleanor Sanderson [1,2,7]

The unexplained protective effect of childhood adiposity on breast cancer risk may be mediated via mammographic density (MD). Here, we investigate a complex relationship between adiposity in childhood and adulthood, puberty onset, MD phenotypes (dense area (DA), non-dense area (NDA), percent density (PD)), and their effects on breast cancer. We use Mendelian randomization (MR) and multivariable MR to estimate the total and direct effects of adiposity and age at menarche on MD phenotypes. Childhood adiposity has a decreasing effect on DA, while adulthood adiposity increases NDA. Later menarche increases DA/PD, but when accounting for childhood adiposity, this effect is attenuated. Next, we examine the effect of MD on breast cancer risk. DA/PD have a risk-increasing effect on breast cancer across all subtypes. The MD SNPs estimates are heterogeneous, and additional analyses suggest that different mechanisms may be linking MD and breast cancer. Finally, we evaluate the role of MD in the protective effect of childhood adiposity on breast cancer. Mediation MR analysis shows that 56% (95% CIs [32%–79%]) of this effect is mediated via DA. Our finding suggests that higher childhood adiposity decreases mammographic DA, subsequently reducing breast cancer risk. Understanding this mechanism is important for identifying potential intervention targets.

Breast cancer is the most common cancer in women worldwide[1]. Incidence rates continue to rise globally[2], and thus there is an urgent need to identify new and modifiable breast cancer risk factors. It is also critical to investigate the links between protective traits and breast cancer as those may reveal new mechanisms for targeted intervention. Observational and Mendelian randomization (MR) studies have shown that adiposity in childhood may reduce the risk of breast cancer in later life[3–7], and that this effect is direct and independent of adult body size. MR is an approach to causal inference that uses genetic variants as instrumental variables (IVs) to infer whether a modifiable health exposure influences a disease outcome[8,9]. In previous work[10], we used an MR framework to investigate the biological mechanism underlying the protective effect of childhood adiposity by reviewing several potential mediators, including hormonal, reproductive, and glycaemic

[1]University of Bristol, MRC Integrative Epidemiology Unit, Bristol, UK. [2]University of Bristol, Population Health Sciences, Bristol, UK. [3]Icahn School of Medicine at Mount Sinai, Department of Genetics and Genomic Sciences, Department of Population Health Science and Policy, New York, NY, USA. [4]University of Texas MD Anderson Cancer Center, Department of Epidemiology, Houston, TX, USA. [5]Kaiser Permanente Northern California, Division of Research, Oakland, CA, USA. [6]University of Bristol, School of Cellular and Molecular Medicine, Bristol, UK. [7]These authors contributed equally: Bethan Lloyd-Lewis, Eleanor Sanderson. ✉e-mail: marina.vabistsevits@bristol.ac.uk

**Fig. 1 | Flow diagram of relationships between traits investigated in this study.** Blue arrows indicate a negative (decreasing/protective) effect and pink arrows show a positive (increasing/causal) effect relationship, as previously reported in the literature. The numbers signpost the analysis sections, which are mentioned throughout the text and correspond to the numbers in the analysis summary in Table 1.

traits. However, none of the investigated mediators sufficiently explained the protective effect of childhood adiposity on breast cancer risk. A mediator that has not yet been thoroughly investigated is mammographic density (MD), an established risk factor for breast cancer[11,12].

MD refers to the radiological appearance of fibroglandular vs adipose tissue in the breast and is frequently quantified in three phenotypes: dense area (fibroglandular tissue, DA), non-dense area (adipose tissue, NDA) and percent density (dense area as a proportion of total breast area, PD). DA and PD are associated with an increased risk of breast cancer, whereas NDA is independently associated with a decreased risk[13]. A high DA and PD elevate breast cancer risk as tumours are more likely to arise in fibrous tissue, as well as being more difficult to detect in dense areas on a mammography exam[14]. MD is highly heritable[15] and the risk of developing cancer is 4-6 fold higher in women with extremely dense vs fatty breasts[14], but MD appears to be similarly associated with all breast cancer molecular subtypes[16,17]. Although the association between MD and breast cancer is well-established, the molecular and cellular events that lead to the development of MD and why it is associated with increased cancer risk are not well understood[18].

Growing evidence points to associations between childhood adiposity, puberty onset, and adult mammographic density (reviewed in ref. 18). Puberty is a critical time for breast development, during which the breast epithelial and stromal compartments undergo extensive growth and tissue remodelling[19]. Later age at menarche has been shown to positively associate with higher MD[20,21], despite being associated with a decreased risk of breast cancer[22,23]. Adiposity at different developmental stages also affects MD, as increased body size in adolescence is associated with a higher abundance of adipose non-dense tissue and lower dense area and percent density in adulthood[18,20,24,25]. Childhood adiposity also has a well-established effect of decreasing age at menarche[26], which in turn leads to higher adult adiposity[27]. Taken together, these traits appear to have a complex and interlinked relationship that impacts breast development and growth, and, ultimately breast cancer risk. Several recent observational studies have suggested that childhood adiposity may confer long-term protection against breast cancer via its effect on mammographic breast density[28–31]. The effect of MD on breast cancer has also been analysed using different MR methods[10,32,33]. While the overall picture reported from these studies supported observationally known associations, there were some differences depending on the MR method employed, suggesting sensitivity to the underlying assumptions.

Here, we explore the mediating role of mammographic density in the protective effect of high childhood adiposity on breast cancer risk, using data from genome-wide association study (GWAS) studies of

childhood body size, adult body size, age at menarche, mammographic density, and breast cancer within a Mendelian randomization framework.

## Results
### Study overview
In this study, we aimed to investigate the mediating role of mammographic density in the protective effect of childhood adiposity on breast cancer risk. Figure 1 presents a flow diagram of the relationships between the investigated traits. The summary of all analyses conducted is presented in Table 1. First, we examined the effect of body size (childhood and adulthood) on mammographic density (dense area, non-dense area, percent density) using univariable MR and multivariable MR (MVMR)[25]. We then reviewed the role of age at menarche in the childhood body size effect on MD phenotypes. Next, using data from the Breast Cancer Association Consortium (BCAC)[34,35] (Supplementary Table 1), we assessed the effect of MD phenotypes on breast cancer risk. We further investigated pleiotropy among the genetic instruments for the MD phenotypes using a variety of advanced sensitivity analysis methods[36–38], PheWAS[39], and pathway analysis, to dissect their heterogeneous effect and improve the understanding of the MD effect on breast cancer. Finally, we performed MVMR of childhood body size and MD phenotypes with breast cancer risk and mediation analysis to assess the direct and indirect effects of both traits and evaluate the role of MD in the poorly understood protective effect of childhood body size on breast cancer.

This study is reported as per the guidelines for strengthening the reporting of Mendelian randomization studies (STROBE-MR)[40,41].

### Body size effect on mammographic density
We used univariable MR to evaluate the total effect of childhood and adult body sizes on each MD phenotype (analysis #1 in Table 1 and Fig. 1). This analysis was performed using MD GWAS data unadjusted for adult BMI to avoid double adjustment for BMI in MVMR analyses; the details of this and subsequent analyses using MD GWAS data adjusted for adult BMI (i.e. the data from the original publication of MD GWAS[32]) are available in Supplementary Note 1.

We found evidence that larger body size, both during childhood and as an adult, reduces dense area (effect size −0.63 [95% CI −0.76: −0.49] and −0.51 [95% CI −0.63: −0.38], respectively) and percent density (−0.88 [95% CI −1.01: −0.74] and −0.95 [95% CI −1.06: −0.83]), but increases non-dense area (0.81 [95% CI 0.67: 0.96] and 1.08 [95% CI 0.97: 1.19]) (Fig. 2a, Supplementary Data 1). The estimates from these analyses reflect the standard deviation (SD) change in MD phenotype for each change in childhood and adult body size category.

We also performed multivariable MR of childhood and adult body size to estimate the direct effects of body size at each age on MD conditional on the other age (Fig. 2b, Supplementary Data 3). In this analysis, a direct effect was demonstrated for both traits, however, larger childhood body size had a stronger effect on decreasing dense area (−0.53 [−0.70: −0.37] vs -0.21 [−0.37: −0.05]), while larger adult body size had a stronger effect on increasing non-dense area (adipose tissue area of the breast) (0.28 [0.12: 0.44] vs 0.93 [0.77: 1.08]). The direct effect on percent density was greater from adult body size, but its magnitude was considerably reduced in MVMR for both measures (−0.49 [−0.66: −0.34] and −0.66 [−0.83: −0.52]).

### Age at menarche effect on mammographic density
In this MR analysis, we sought to analyse childhood body size and age at menarche together to evaluate their total and direct effects on MD phenotypes (analysis #2 in Table 1 and Fig. 1). In univariable MR (Fig. 2a, Supplementary Data 5), childhood body size and age at menarche had strong opposing effects on MD (age at menarche effect on DA, effect size 0.15 [95% CIs 0.05: 0.26], and PD 0.24 [0.13: 0.35]), which is in agreement with published studies[20,21,25]. In MVMR (Fig. 2c,

**Table 1 | Summary of analyses conducted**

| Analysis type | Exposure trait(s) represented as genetic instruments | Phenotypic outcome traits(s) (when applicable) | Results available in |
|---|---|---|---|
| *Analysis #1* | | | |
| MR | Childhood body size | Mammographic density (DA, NDA, PD) | Fig. 2a, Supplementary Data 1 |
| MR | Adult body size | Mammographic density (DA, NDA, PD) | Fig. 2a, Supplementary Data 1 |
| MVMR | Childhood body size, Adult body size | Mammographic density (DA, NDA, PD) | Fig. 2b, Supplementary Data 3 |
| *Analysis #2* | | | |
| MR | Age at menarche | Mammographic density (DA, NDA, PD) | Fig. 2a, Supplementary Data 5 |
| MVMR | Childhood body size, age at menarche | Mammographic density (DA, NDA, PD) | Fig. 2c, Supplementary Data 7 |
| *Analysis #3* | | | |
| MR | Dense area (DA) | Breast cancer (overall and subtypes) | Fig. 3a, Supplementary Data 9 |
| MR | Non-dense area (NDA) | Breast cancer (overall and subtypes) | Fig. 3a, Supplementary Data 9 |
| MR | Percent density (PD) | Breast cancer (overall and subtypes) | Fig. 3a, Supplementary Data 9 |
| *Analysis #4* | | | |
| MVMR | Childhood body size, Dense area (DA) | Breast cancer (overall and subtypes) | Fig. 3b, Supplementary Data 11 |
| MVMR | Childhood body size, Non-dense area (NDA) | Breast cancer (overall and subtypes) | Fig. 3b, Supplementary Data 11 |
| MVMR | Childhood body size, Percent density (PD) | Breast cancer (overall and subtypes) | Fig. 3b, Supplementary Data 11 |
| *Analysis #5* | | | |
| MR-PRESSO | Mammographic density (DA, NDA, PD) | Breast cancer overall sample | Fig. 4a, Supplementary Figs. 4a, 6a, Supplementary Data 14 |
| Radial-MR | Mammographic density (DA, NDA, PD) | Breast cancer overall sample | Fig. 4b, Supplementary Figs. 4b, 6b, Supplementary Data 15 |
| MR-Clust | Mammographic density (DA, NDA, PD) | Breast cancer overall sample | Fig. 4c, d, Supplementary Figs. 4c, d, 6c, d, Supplementary Data 16 |
| *Analysis #6* | | | |
| PheWAS | Mammographic density (DA, NDA, PD) | N/A | Fig. 5, Supplementary Figs. 5, 7 Supplementary Data 17–19 |
| *Analysis #7* | | | |
| Pathway analysis | Mammographic density (DA, NDA, PD) | N/A | Supplementary Data 20–25 |
| *Analysis #8* | | | |
| Mediation analysis | Childhood body size, Dense area (DA) (as a mediator) | Breast cancer overall sample | Supplementary Note 2 |

The table is split into analysis sections (#) for convenient reference throughout the text. Mammographic density (MD) is available as three phenotypes: Dense area (DA), non-dense area (NDA), and percent density (PD); data source: Sieh et al.[32] Breast cancer outcomes include data from BCAC 2017 and 2020 (overall samples, ER + /ER- samples and five molecular subtypes: Luminal A, Luminal B1 (HER2 + ), Luminal B2 (HER2-), HER2-enriched, and triple-negative; summarised in Supplementary Table 1; data sources[34,35]:). Childhood/adult size body and age at menarche data are UK Biobank phenotypes from Richardson et al.[4] (female-only data, including for instrument extraction). In the table, when several exposures/outcomes are listed (e.g. MD phenotypes or cancer subtypes), this indicates that MR analysis was done separately for each, unless there are two exposures in MVMR. *MR* Mendelian randomization, *MVMR* Multivariable MR, *BCAC* Breast Cancer Association Consortium.

Supplementary Data 7), the direct effect of body size on DA conditional on age at menarche is similar to the total effect (−0.66 [−0.82: −0.51]), while the effect of age at menarche on DA is attenuated to overlap the null (−0.06 [−0.18: 0.05]). Adiposity in childhood reduces MD and lowers the age at menarche (as shown in[10]), while younger age at menarche has a negative effect on MD (i.e. the inverse of higher age at menarche increasing MD in Fig. 2a). The attenuation of age at menarche effect can be explained in the following way: (1) the direct effect of childhood adiposity is maintained in MVMR when accounting for age at menarche, suggesting that adiposity affects MD independently of starting puberty earlier, (2) the menarche effect in univariable results is not present in MVMR results suggesting that it is largely due to unaccounted increased childhood adiposity (and its effect on the initiation of puberty). Collectively, our results show that the density-decreasing effect of larger childhood body size is not acting via lowering the age at menarche, and that childhood body size and age at menarche may have entirely different mechanisms linking them to breast cancer.

## Mammographic density effect on breast cancer

Next, we evaluated the effect of BMI-unadjusted MD phenotypes on breast cancer (analysis #3 in Table 1 and Fig. 1) using IVW MR estimation. The total effect of MD phenotypes on breast cancer subtypes is presented in Fig. 3a (Supplementary Data 9). Overall, we found a consistent trend in the direction of effect across all breast cancer subtypes for each MD exposure trait: dense area and percent density increased the risk, while non-dense area decreased the risk, which is in line with the observational data. Despite being consistent, many

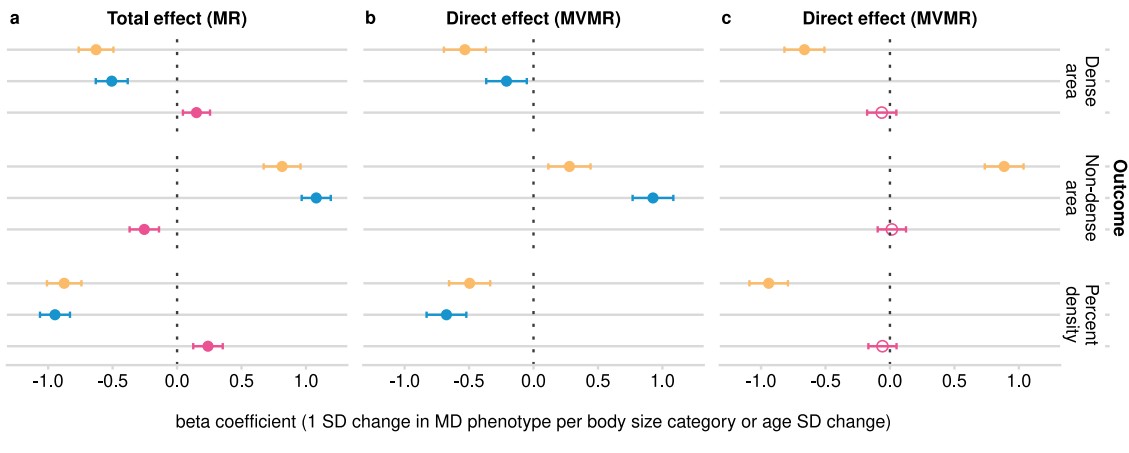

**Fig. 2 | The effect of childhood body size, adult body size, and age at menarche on MD phenotypes (dense area, non-dense area, percent density; unadjusted for BMI at GWAS level). a** Total effect of each exposure trait on MD outcomes (univariable MR). **b** Direct effects of childhood and adult body sizes on MD outcomes (MVMR). **c** Direct effects of childhood body size and age at menarche on MD outcomes (MVMR). The effect is reported as the standard deviation (SD) change in MD phenotype per body size category or age at menarche SD change. The error bars indicate 95% confidence intervals around the point estimate (beta coefficient) from IVW MR and IVW-MVMR analyses. The empty circle data points highlight the results where confidence intervals overlap the null. *MR* Mendelian randomization, *MVMR* multivariable MR. GWAS sample sizes: childhood/adult body size: $n = 246,511$; age at menarche: $n = 143,819$; MD phenotypes: $n = 24,158$. Source data are provided as a Source Data file.

estimates were imprecise, however, there was stronger evidence for a positive effect of dense area on overall breast cancer (OR 1.38 [95% CI 1.002: 1.90]), ER+ breast cancer, and several subtypes. The individual SNP-specific effects within all MD phenotypes' total estimates were heterogeneous (detailed below under **Sensitivity analysis**), and therefore, in the **Mammographic density genetic instruments investigation** section we explore those effects using various sensitivity and outlier detection methods. The direct effects from the MVMR analysis in Fig. 3b are discussed in a later section.

## Sensitivity analysis

To investigate potential violations of the MR assumptions and validate the robustness of the two-sample IVW MR results, we performed additional MR analyses using MR-Egger[42] and weighted median[43] approaches, both of which provide sensitivity analyses that are more robust to particular forms of horizontal pleiotropy. The Egger intercept was used to explore the potential for the presence of directional horizontal pleiotropy, and Cochran's Q statistic[44] was calculated to quantify the extent of heterogeneity among SNPs, which is indicative of potential pleiotropy. For MVMR, we tested instrument strength, using a conditional F-statistic[45] and examined heterogeneity using an adapted version of the Q-statistic ($Q_A$).

The estimated total effects of childhood and adult body size measures on MD phenotypes were consistent across MR sensitivity analyses with Egger intercept 0.01 or lower. The F-statistics were >10 and Q-statistics did not indicate excessive heterogeneity (Supplementary Data 2). In MVMR, the conditional F-statistics were also above 10, indicating that weak instrument bias is unlikely to be present[45]. The presence of directional pleiotropy was assessed by estimating $Q_A$ statistics, which also were not notably large (Supplementary Data 4).

The direction of effect was consistent among the MR methods when assessing age at menarche effect on MD phenotypes, but there was less robust evidence of effect in the weighted median result. The F-statistic for age at menarche was above 10; the Egger intercept was substantially close to zero (-0.002), indicating little evidence of directional pleiotropy[46]. The Cochran's Q value was large with p-values < 2 x 10⁻¹⁰, indicating high heterogeneity (Supplementary Data 6). In MVMR of age at menarche and childhood body size, the F-statistics

were above 10, and $Q_A$ was similar to the Q value in the univariable analysis (Supplementary Data 8).

In the main IVW analysis of MD phenotypes' effect on breast cancer outcomes, the evidence was present only in selected exposure-outcome pairs, as described in the previous section. Applying sensitivity methods to those results showed some inconsistency, with MR-Egger producing imprecise results. The weighted median approach, which relies on at least 50% of the variants' total weight being from valid instruments[43], provided evidence for an effect in substantially more analyses than IVW, which relies on 100% of variants being valid instruments, indicating that some variants may be outliers (Supplementary Data 9). The Egger intercept in the analyses of non-dense area and percent density with subtype outcomes suggested the likely presence of horizontal pleiotropy. The intercept in analyses of dense area, where evidence of effect was present in IVW, was smaller, indicating that dense area phenotype is less subject to pleiotropy. The MD phenotypes' instrument strength was good (F-statistics >10), suggesting that weak instruments are unlikely to be a source of serious bias in the univariable analysis. Steiger filtering did not indicate that MD phenotypes' instruments explained more variance ($R^2$) in breast cancer rather than in MD phenotypes, and therefore, were not excluded from the analysis. Interestingly, we identified substantial heterogeneity for all MD phenotypes, suggested by very high Q-values with small p-values. High heterogeneity may be indicative of one or more variant outliers in the analysis, which was explored with additional sensitivity in the next section. The sensitivity analysis details are available in Supplementary Data 10.

## Mammographic density genetic instruments investigation

To explore the high heterogeneity in the genetic instruments for the MD phenotypes, we applied several methods that aim to dissect heterogeneity and assess potential horizontal pleiotropy through outlier detection (analysis #5 in Table 1). In this investigation, we focused on the overall breast cancer sample outcome as the main analysis, but additional results for breast cancer subtypes are available in Supplementary Note 3 (Supplementary Fig. 10a–i).

We used MR-PRESSO[36] and Radial-MR[38] (see **Methods**) to identify the variant outliers (Supplementary Data 14, 15). For dense area, both methods determined the same set of SNPs as outliers (Fig. 4a, b). The

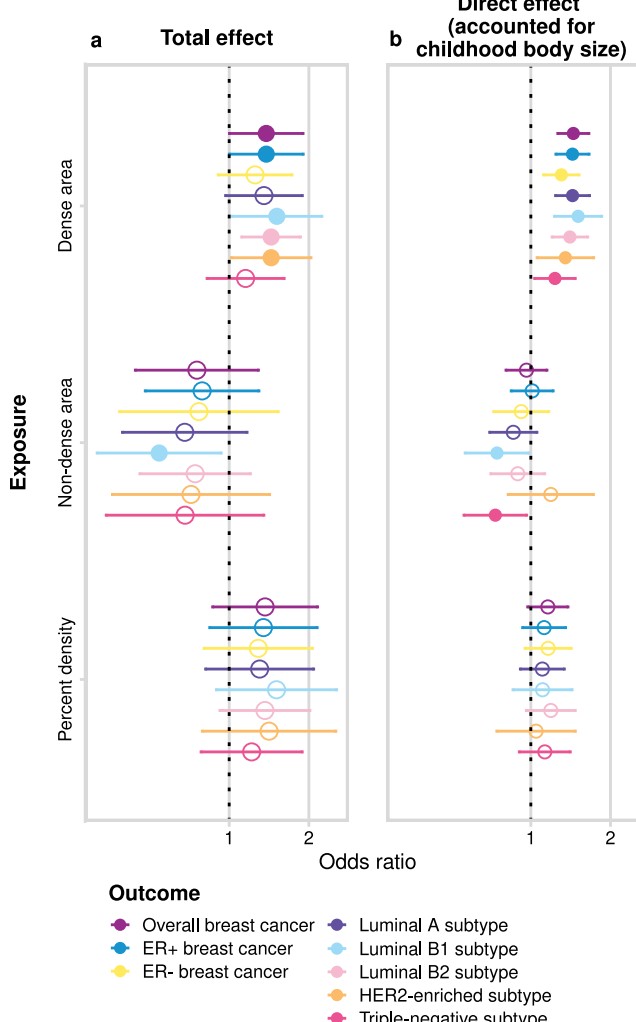

**Fig. 3 | The effect of MD phenotypes (dense area, non-dense area, percent density; unadjusted for BMI at GWAS level) on breast cancer (overall sample from BCAC 2017 and subtype samples). a** The total effect (univariable MR). **b** The direct effect (accounted for childhood body size, MVMR). The plots show the odds of breast cancer per SD increment in MD phenotype. The error bars indicate 95% confidence intervals around the point estimate (odd ratio) from IVW MR and IVW-MVMR analyses. The empty circle data points highlight the results where confidence intervals overlap the null. GWAS sample sizes: MD phenotypes: $n = 24,158$; breast cancer overall sample: $n = 247,173$ (133,384 cases and 113,789 controls); breast cancer subtypes sample sizes and details are provided in Supplementary Table 1. Source data are provided as a Source data file.

outlier-corrected total IVW estimates are presented below the single SNP forest plots (outlier SNPs are highlighted), alongside the results of other MR methods. With outliers removed, the point estimate (OR 1.40 [1.26: 1.56]) is similar to the original IVW result (OR 1.38 [1.002: 1.90]), but the confidence intervals are more precise. Consequently, the outlier-corrected IVW estimates of dense area had stronger evidence of effect on breast cancer, and were similar to weighted median method results (OR 1.25 [1.12: 1.39]).

Next, we used MR-Clust[37] to investigate the presence of clustered heterogeneity among the genetic variants. MR-Clust groups genetic variants into clusters with similar estimates for the causal effect of the exposure on the outcome (i.e. based on their direction, magnitude, and precision). A cluster may represent a distinct pathway through which exposure is related to the outcome, and investigating heterogeneous estimates in this way may reveal additional information about the exposure-outcome relationship (see Methods for further details).

MR-Clust detected three distinct clusters ('cluster_1', 'cluster_2', 'cluster_3'), a 'null' cluster, and two 'junk' SNPs that were not assigned to any of the clusters (Fig. 4c, Supplementary Data 16). We see that the heterogeneity outliers flagged by MR-PRESSO and Radial-MR (Fig. 4a, b) represent separate clusters in MR-Clust (Fig. 4d). 'Cluster_2' (blue) is equivalent to the outlier-corrected estimate from those earlier analyses and the variants in this cluster are positively associated with an increase in both dense area and breast cancer risk. 'Cluster_3' (orange) and a positive 'junk' SNP are associated with breast cancer to a higher magnitude (Fig. 4c) and therefore form a separate cluster. Interestingly, the SNPs in 'cluster_1' are protective of breast cancer despite being associated with increased density. It is important to note that both the inverse association ('cluster_1') and the same direction but higher magnitude association ('cluster_3') clusters add to the overall heterogeneity of the total estimate.

The results for non-dense area and percent density phenotypes are presented in Supplementary Figs. 4 and 6. We similarly found outliers and clusters in those traits' instruments. However, due to the lower number of instruments available for these traits, the results from MR-PRESSO and Radial-MR should not be overinterpreted. The outlier-corrected IVW estimates (non-dense area – OR 0.75 [0.65: 0.86] and percent density – OR 1.29 [1.16: 1.44]) were similar to the weighted median method results (OR 0.74 [0.63: 0.87] and OR 1.32 [1.14: 1.53], respectively) (Supplementary Data 14 and 15). In MR-Clust, for non-dense area and percent density, there were also variants that associated with breast cancer in the opposite direction to the overall and expected effect from the exposure (e.g. negatively associated with breast cancer risk but positively associated with a factor causal for breast cancer, or vice versa) – two 'negative effect' outliers for percent density and one 'positive effect' outlier for non-dense area).

## PheWAS analysis
We carried out a phenome-wide association study (PheWAS) analysis[39] on the genetic instruments for the MD phenotypes to examine their associations with other traits (analysis #6 in Table 1). We aimed to review the differences between associations by clusters identified with MR-Clust and evaluate whether outlier SNPs may be strongly associated with other phenotypes, which may explain the horizontal pleiotropic effect and hint at alternative causal pathways for those outliers.

The PheWAS results for the dense area phenotype are plotted in Fig. 5. The SNPs that were identified as outliers in previous analyses and that formed distinct clusters from the main effect clusters, have a higher number of associations with other traits, highlighting their pleiotropic effect. In the plot, we use the diamond shape to indicate dense area SNPs that associate strongly ($p$-value $< 5 \times 10^{-8}$) with breast cancer. Those SNPs correspond to 'cluster_3', 'cluster_1', and 'junk' cluster SNPs in the MR-Clust results, here similarly flagging their association with breast cancer risk, which may be happening via a different pathway other than through dense area.

PheWAS plots for non-dense area and percent density are available in Supplementary Figs. 5 and 7. For those phenotypes, similarly, we found associations with breast cancer for the outlier SNPs. All found associations are available in Supplementary Data 17–19.

## Gene and pathway overview
To gain some biological context for the identified outlier SNPs and distinct clusters among the MD instruments, we mapped the variants used in MR analyses to genes (Supplementary Data 20–22; see gene-labelled forest plots in Supplementary Fig. 8) and identified pathways that those genes are involved in (analysis #7 in Table 1). Performing a formal gene-set enrichment analysis was not possible here due to the limited number of SNPs available for each phenotype/cluster. Therefore, instead, we created a simple overview of pathway sets that came up for genes in positive and negative effect clusters (Supplementary Data 23–25, Supplementary Fig. 9).

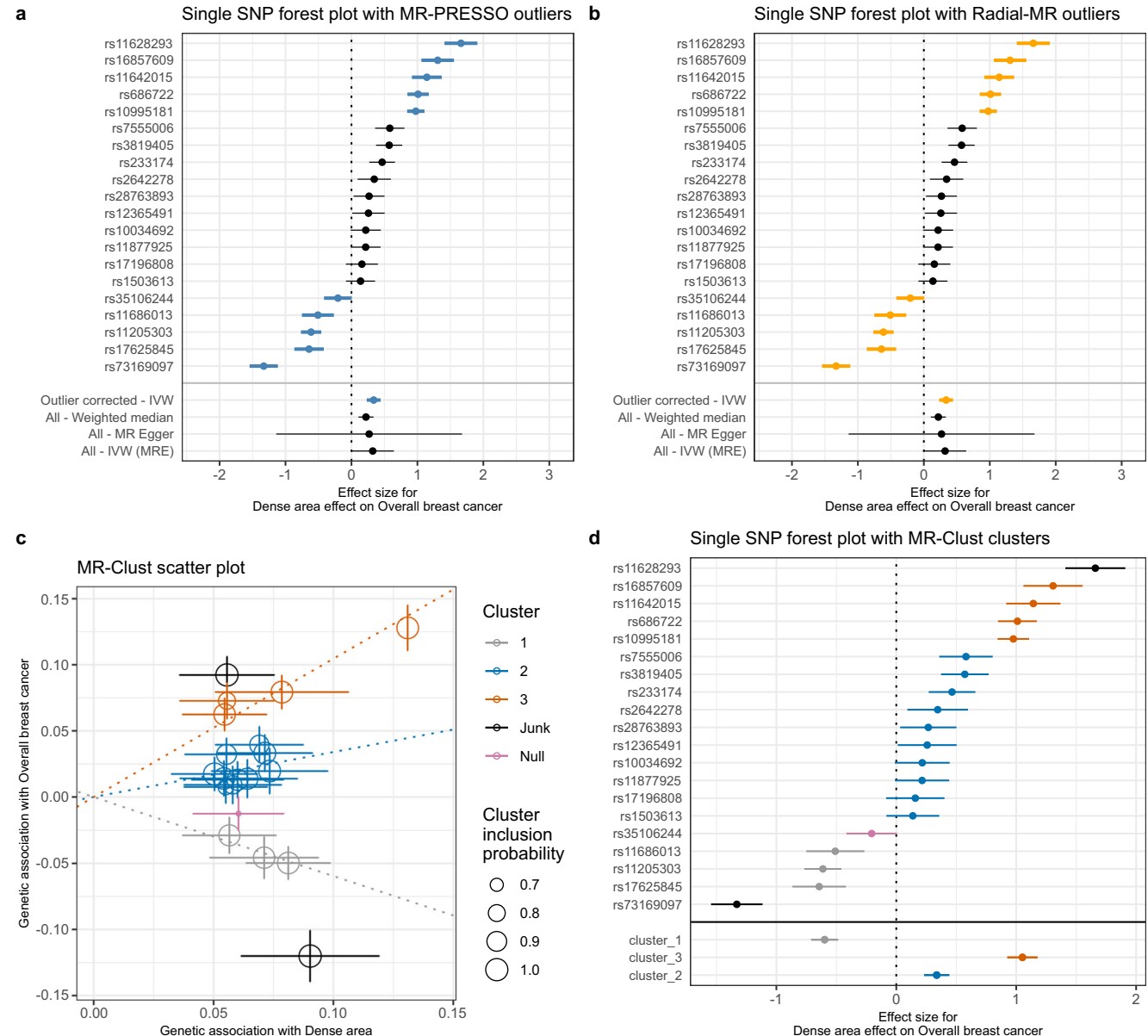

**Fig. 4 | Exploring the heterogeneity of genetic instruments of dense area phenotype on overall breast cancer (BCAC 2017). a** Single SNP forest plot (Wald Ratio estimates), with SNPs identified as outliers by MR-PRESSO marked bold and coloured blue. The outlier corrected estimate is presented along with the standard MR methods estimates. **b** Single SNP forest plot with SNPs identified as outliers by Radial-MR marked bold and coloured yellow. The outlier corrected estimate is presented along with the standard MR methods estimates. **c** MR-Clust scatter plot showing genetic association with dense area and breast cancer per SD change in dense area. Each genetic variant is represented by a point. Colours represent the clusters, and dotted lines represent the cluster means, the point size denotes cluster inclusion probability. The "null" cluster, coloured pink, relates to variants with null effect, whilst the black "junk" cluster are variants that were not assigned to any cluster. **d** Single SNP forest plot with SNPs coloured by the cluster membership assigned by MR-Clust (using the same colours as in the scatter plot). The IVW MR estimates for each cluster are presented below single SNP estimates. In panel **c**, the error bars denote 95% confidence intervals for the genetic associations for each variant. In panels **a**, **b**, **d**, the error bars are 95% confidence intervals of the Wald Ratio point estimate (beta coeficient) for each variant. The effect estimates are reported per SD change in dense area phenotype. *IVW* inverse-variance weighted; *MRE* multiplicative random effects; GWAS sample sizes: Dense area: $n = 24{,}158$; breast cancer overall sample: $n = 247{,}173$ (133,384 cases and 113,789 controls). Source data are provided as a Source Data file.

For dense area, we found a number of unique pathways that only appeared in genes/SNPs with a negative effect. Among those genes, most were described in the functional analyses of previously published MD GWAS[32,47,48], such as *MKL1/MRTFA* (rs73169097 – negative 'null' cluster SNP) and *MTMR11* (rs11205303), both of which have dense phenotype-increasing effect but are protective against breast cancer. The potential tumour-inhibiting and tumour-promoting role of *MKL1* was previously acknowledged in ref. 47. *MTMR11* is negatively associated with both dense area and percent density (but as a result of LD

clumping it is an instrument only for dense area). It appears to be involved in phosphoinositides/phosphatidylinositol metabolism pathways, which are also implicated in cancer. For percent density, the genes in negative clusters were also previously described in published functional analyses—*OTUD7B* (rs12048493) and *ZNF703* (rs4286946)[48,49]. Interestingly, the positive outlier in non-dense area instruments is also mapped to *ZNF703* (rs75772194), which is also associated with breast size[49]. The complete overview of cluster/genes/pathways is available in Supplementary Data 23–25.

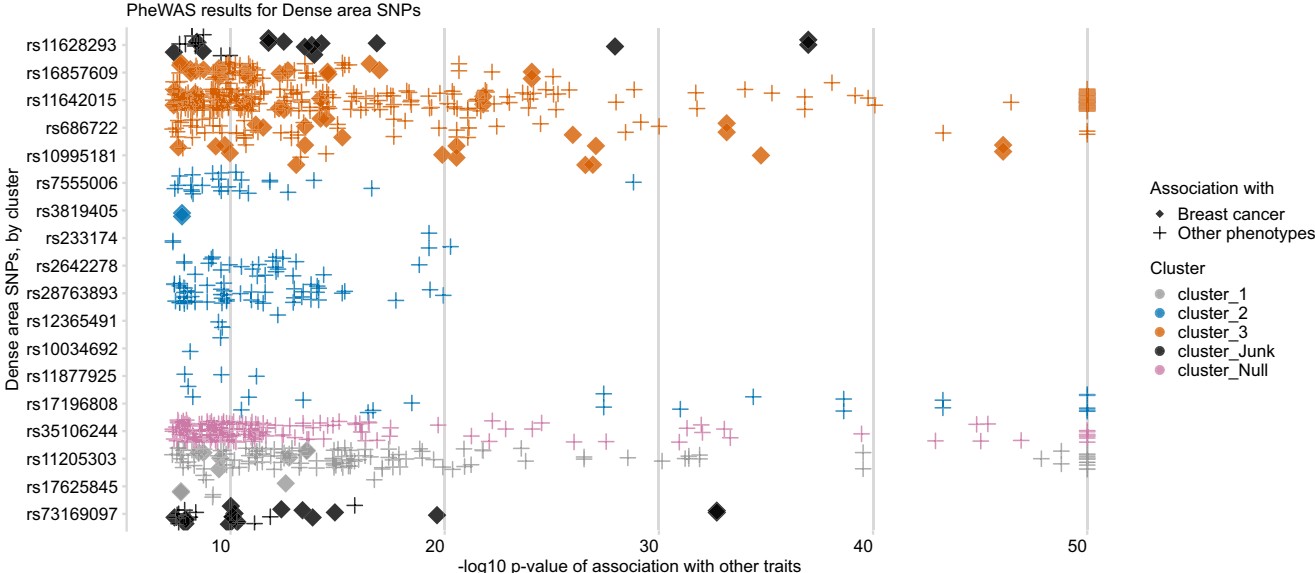

**Fig. 5 | PheWAS results for dense area phenotype genetic variants, ordered by SNP effect and cluster membership (from MR-Clust).** The data points are other traits associated with dense area SNPs (y-axis) at *p*-value (FDR-adjusted) < 5 x 10$^{-8}$ (x-axis, -log10 scale, capped at value 50). The colour shows the cluster membership, in the same palette and order as in Fig. 4c/d. Data points represented by the solid '*diamond*' shapes are breast cancer outcomes; the '*plus*' shapes are all other traits. Source data are provided as a Source Data file.

## Direct effects of mammographic density and childhood body size

In the earlier sections, we reviewed the total effect of MD phenotypes on breast cancer risk (Fig. 3a) and explored it using various sensitivity analyses. In this section, we dissect the direct effects of childhood body size and MD phenotypes on breast cancer risk using MVMR (analysis #4 in Table 1 and Fig. 1). In Fig. 3b (Supplementary Data 11), we see the direct effect of MD on breast cancer accounting for childhood body size, presented alongside the total effect for comparison. There is evidence of a positive direct effect from the dense area on all breast cancer subtypes. The point estimates are similar to those of the total effect, but with more precise confidence intervals. There is evidence of a negative effect from non-dense area on Luminal B1 and triple-negative subtypes, while the effects on other samples have been further attenuated towards the null. For percent density, the magnitude of effect and the uncertainty around the point estimate is reduced in MVMR analysis, with little evidence for an effect of PD on all breast cancer subtypes when accounting for childhood body size. It should be noted that IVW MVMR estimates may also be potentially biased by pleiotropy in the same way as total effect estimates in univariable MR.

From the same MVMR analysis as the results in Fig. 3b, we have also estimated the direct effect of childhood body size on breast cancer accounted for MD phenotypes. Figure 6 presents the total effect of childhood body size on breast cancer (overall and subtypes) (Supplementary Data 13) along with the direct effect accounted for each MD phenotype (Supplementary Data 11). The total effect is strongly protective against all outcomes. In previous work, this protective effect was not disrupted by accounting for any hypothesised mediators[10]. In this analysis, we see that accounting for MD phenotypes attenuates the protective effect making the confidence intervals overlap the null, suggesting that MD may have a role in partially explaining it. When accounting for the dense area, the effect attenuation is seen for all outcomes except the ER- sample. For percent density, the effect on breast subtypes is attenuated but to a lesser extent, which may suggest that dense area phenotype has a stronger mediating role than percent density. For non-dense area, the effect is attenuated also on a subset of breast cancer subtypes. Interestingly,

the effect on ER- subtype is the least affected, suggesting there might be some difference in how MD affects ER- breast cancer risk.

It is important to note that the number of MD instruments in this MVMR analysis was limited (Supplementary Table 2). These MVMR results are also affected by weak instrument bias, as F-statistics are low in these analyses: childhood body size and dense area (F-stat, 17 and 7, respectively) non-dense area (6 and 3), percent density (7 and 4), respectively (Supplementary Data 12).

## Mediation analysis

We performed mediation analysis using MR and MVMR results to assess the role of mammographic density (specifically, dense area) in the relationship between childhood body size and breast cancer. This investigation was also done focusing only on the overall breast cancer sample (analysis #8 in Table 1).

We estimated the indirect effect via MD, using both Product and Difference methods for mediation analysis (see **Methods**). Both methods produced similar indirect point estimates in the same direction, −0.23 [95% CIs −0.33: −0.13] and −0.22 [−0.48: 0:05], respectively. The proportion of the mediated effect via dense area using the Product method estimate was 0.56, indicating that dense area may account for 56% [95% CIs 32% – 79%] of the childhood body size protective effect on breast cancer (see Supplementary Note 2.1 for mediation analysis calculations).

As DA instruments were found to be heterogeneous, we also performed mediation analysis using outlier-corrected estimates to assess the validity of the proportion-mediated effect. The heterogeneity-corrected estimate falls into the CIs of the main results and is more precise (see Supplementary Note 2.2).

## Discussion

The protective effect of higher childhood adiposity on breast cancer risk has been reported in both observational and MR studies[3–7]. However, the mechanism behind this effect has been challenging to decipher, even after reviewing nearly 20 potential mediators[10]. A few observational studies have suggested that mammographic density may have a role in this relationship[28–31]. In this study, we explored the mediating role of MD in the protective effect of higher childhood adiposity

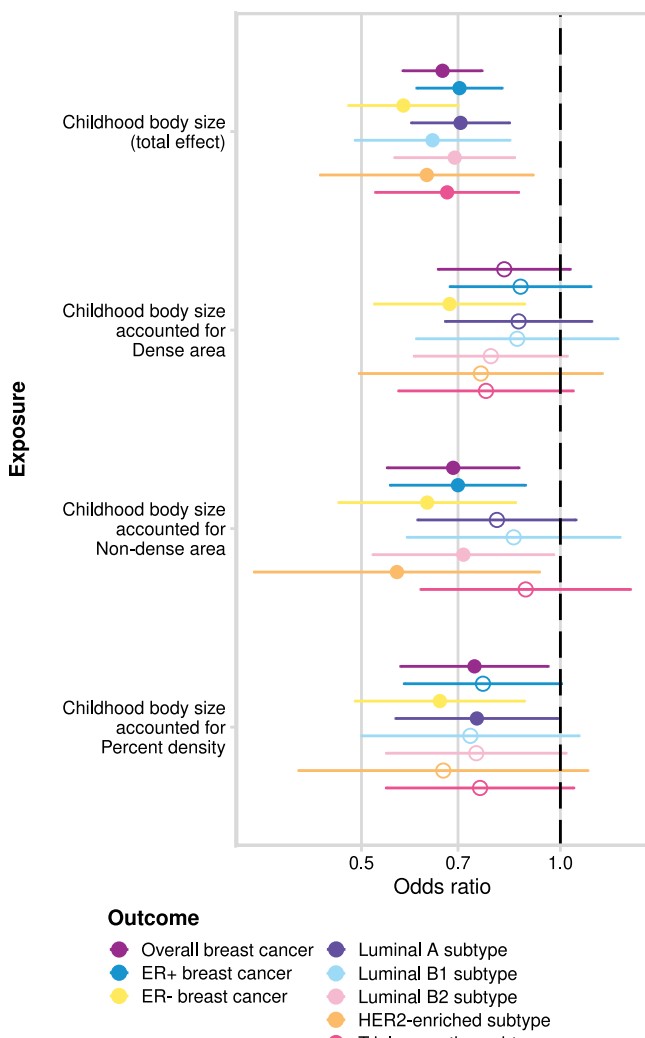

**Fig. 6 | The total effect of childhood body size and the direct effect of childhood body size accounted for MD phenotypes (dense area, non-dense area, percent density; unadjusted for BMI at GWAS level) on breast cancer (overall sample from BCAC 2017 and subtype samples).** The plots show the odds of breast cancer per body size category change. The error bars indicate 95% confidence intervals around the point estimate (odds ratio) from IVW MR and IVW-MVMR analyses. The empty circle data points highlight the results where confidence intervals overlap the null. GWAS sample sizes: MD phenotypes: n = 24,158; breast cancer overall sample: n = 247,173 (133,384 cases and 113,789 controls); breast cancer subtypes samples sizes and details are provided in Supplementary Table 1. Source data are provided as a Source Data file.

on breast cancer risk using Mendelian randomization, examining the complex relationships between childhood body size, adult body size, age at menarche, mammographic density, and breast cancer risk.

Firstly, we investigated the factors that may affect MD – adiposity at different life stages and age at menarche. We found that higher childhood and higher adulthood adiposity decrease dense area and percent density, while both increase the non-dense (adipose tissue) area. In multivariable MR analysis, however, the independent direct effect of childhood adiposity was stronger for decreasing dense area, while adult adiposity was stronger for increasing non-dense area. The inverse effect of higher body size on density is likely explained by increasing breast adiposity, reducing the proportion of fibroglandular components, and increasing adipocyte differentiation of stromal cells, thus reducing collagen production[50]. As breast tissue undergoes substantial development during puberty, it is reasonable that childhood rather than adult adiposity is a more important factor for dense area.

The stronger effect of adult adiposity on the non-dense area is likewise logical, as the change in MD with age is reflected in glandular tissue reduction and an increase in fat[51]. We also showed that adjustment for BMI in GWAS may lead to an unexpected and misleading result in MR analysis (Supplementary Note 1), if BMI (i.e., heritable covariate) also has a role in the studied relationship[52,53].

The previously observed association of age at menarche with breast density[20] was replicated in our MR analysis, with later menarche increasing dense area and percent density and decreasing non-dense area. In MVMR with childhood body size, however, the effect of age at menarche on MD phenotypes was attenuated. Greater adiposity in childhood reduces dense area and percent density and lowers the age at menarche[10], while earlier menarche decreases dense area and percent density. Therefore, the attenuation of its effect in MVMR indicates that the menarche effect observed in the univariable analysis may be due in part to increased adiposity (and its effect on the initiation of puberty), as earlier suggested[54]. Overall, our results suggest that the density-decreasing effect of childhood body size is not acting predominantly via lowering the age at menarche.

This finding draws attention to prior MR studies showing little evidence of effect of age at menarche on breast cancer risk[10,55]. Interestingly, in MVMR analyses when accounting for BMI, there is a shift from the neutral effect to a causal effect with earlier age at menarche increasing the risk. It is likely that the total effect of age at menarche is driven (and disguised) by childhood BMI SNPs in the age at menarche GWAS instruments, and accounting for BMI in MVMR separates the independent effects of childhood BMI and age at menarche on breast cancer risk. Taken together with our finding that MD is not affected by age at menarche when accounting for body size, this suggests that the mechanisms linking childhood adiposity and age at menarche to breast cancer could be entirely different and operate in opposite directions. Uncovering the mechanistic links in both relationships (as partly done in this work with respect to childhood body size) will identify different pathways that could be modifiable and, together, could contribute a very substantial component of modifiable breast cancer risk. Another important consideration relating to mechanistic links is the distinction between mutagenesis and promoters in breast cancer causation[56], which may also contribute to the differential effects of childhood adiposity and age at menarche on breast cancer risk.

The central relationship explored in our study is that of MD and breast cancer, and whether MD helps explain the inverse association of childhood adiposity and breast cancer risk. When examining the total effect of MD phenotypes on breast cancer risk (overall and subtypes), we observed consistent trends in the direction of effects, with dense area and percent density increasing the risk and non-dense area decreasing the risk, in line with observational results[11,13]. We found evidence of a positive effect from the dense area on breast cancer risk overall and for certain subtypes, but for other MD exposure/breast cancer outcome pairs the evidence was insufficient. The results produced by the IVW method may potentially be biased by pleiotropy, therefore the detected high levels of heterogeneity were further explored in our analysis and will be discussed below. It is also worth mentioning that our results may slightly differ from the previously published MR results using related data[10,32,33], which could be explained by the differences in the MR methods employed, the approach to instrument selection, and the fact that the MD GWAS was unadjusted for BMI in this study.

While the total effect of MD on breast cancer was imprecisely estimated, IVW-MVMR of MD phenotypes with childhood body size showed strong evidence of a risk-increasing direct effect from the dense area on all breast cancer subtypes, with less evidence for a negative effect of non-dense area and a lack of evidence for an effect of percent density. This highlighted the possibility that dense area is the more important risk factor for breast cancer, however in observational

studies[11], percent density has been found to have a stronger association because it combines the effects of both dense area and non-dense area which have distinct genetic aetiologies[32]. The direct effect of childhood body size on breast cancer was attenuated in this MVMR analysis, suggesting a potential mediating role of mammographic density in the relationship between them. We have not observed such attenuation of the effect of childhood body size in our previous work, where many potential mediations were assessed, with the body size effect remaining unaffected[10]. Interestingly, this attenuation of effect was not present in analyses of ER-negative breast cancer, suggesting that there might be some differences in how MD affects this disease subtype. We considered including adult body size and age at menarche as covariates in MVMR, however, we opted not to pursue this analysis due to concerns about the statistical power.

In addition to the effect changes observed in MVMR, we conducted a formal mediation analysis with the dense area phenotype. Both mediation methods we applied produced very similar indirect effect estimates (−0.23 and −0.22, Product and Difference methods, respectively). Such agreement of estimates was not the case for other mediators we reviewed in our previous work[10]. The confidence intervals around these estimates were more precise for the Product method −0.23 [−0.33: −0.13]). The calculated proportion mediated via dense area suggested that 56% [32% − 79%] of childhood adiposity's protective effect could be due to it decreasing the dense area in childhood, which leads to reduced breast cancer risk in adulthood.

The above finding is promising, however, the relationship of MD phenotypes with breast cancer is complex and, as shown in our sensitivity analyses, the genetic variants used in the analysis have heterogeneous estimates and are potentially highly pleiotropic. We thoroughly evaluated the dense area, non-dense area, and percent density genetic instruments using several robust MR outlier detection methods and the MR clustering method, MR-Clust, to decompose heterogeneity in the results. For dense area and percent density, we found a set of outlier SNPs that together formed 'negative effect' clusters, which mapped to genes that were associated with higher dense area/percent density, but a decreased cancer risk. This has been previously reported for the same identified genes, e.g. *MLK1* in ref. 47 and *MTMR11* in ref. 48. Similarly, for the non-dense area, we found one SNP with the opposite effect on breast cancer to the overall effect direction. The PheWAS analysis highlighted the fact that outlier SNPs, which also form separate clusters of MD effect on breast cancer, were highly pleiotropic, with the majority also associated with breast cancer. Several methods for outlier correction showed that removing those SNPs results in stronger and more consistent effects of MD phenotypes on breast cancer risk.

The discovery of multiple MD variants that are also breast cancer susceptibility loci, highlights their shared genetic component and the critical role MD plays as an intermediate phenotype for the disease. The inconsistency in the direction of associations between some MD-associated SNPs and breast cancer risk is perplexing, and is the reason for the observed heterogeneity in MR estimates. One potential explanation for discrepancies in these variants may be that multiple alternative pathways are involved, and are acting across different life stages, which differentially affect breast development and the risk of breast cancer. There is also a strong possibility that not all contributors to MD influence breast cancer risk. Understanding, and correctly classifying the driving components of MD (reviewed in ref. 57) into those that influence breast cancer risk, and using those for future studies could increase results precision and the degree of mediation detected. Motivated by a recent study that explored a similarly heterogenous effect of IGF-1 on type 2 diabetes using MR-Clust and pathway analysis[58], in our work, we attempted to characterise pathways that may be underlying the identified positive and negative effect clusters. In our case, however, due to the limited number of instruments, pathway gene-set enrichment analysis was not feasible. An extensive pathway analysis based on the MD GWAS used in our work was reported in the original publication[32].

The limitations of our study, including the precision of estimates and pathway analysis, can be attributed to the small sample size of the currently available MD GWAS data and the consequent low number of robustly associated genetic instruments. Despite using one of the largest MD GWAS cohorts to date ($N = 24{,}192$)[32], the number of instruments was still relatively small (albeit higher than in earlier studies, such as[47,59]). A summary table of all published MD GWAS studies is provided in a recent review[60]. A similarly sized MD GWAS conducted on data from the BCAC cohort ($N = 24{,}579$–$27{,}900$)[61] has recently been released, but due to the unavailability of effect sizes, it is not possible to validate our findings using this resource. Once larger MD GWAS studies become available, and more SNPs with robust associations are identified, our results could be replicated. A larger sample size may also allow for a menopause-status stratified analysis, as MD declines with age[62], and the association between density and breast cancer risk is stronger pre-menopause[63]. A higher number of MD instruments would also enable more informative clustering and pathway analyses, despite the likely maintained heterogeneity amongst individual estimates. Furthermore, the estimation of childhood body size indirect effect via MD would also likely be more precise.

It is important to highlight a few recent developments in studying the genetics of mammographic density. Firstly, the first-ever GWAS of breast tissue structure patterns (also referred to as texture features) has recently been published[64], which is an emerging independent breast cancer risk factor[65]. Texture variation can differ substantially between women, despite having the same percent density. Including this trait in the MD phenotype analyses (including MR) can produce additional insights into the development of breast cancer. Secondly, as exploring proximal molecular mediators is becoming more widespread, the analysis of MD phenotypes in the BCAC cohort[61] also included a transcriptome-wide association study (TWAS). The study revealed additional novel associations between imputed breast tissue expression level and MD phenotypes. Some of the identified genes were located in proximity to GWAS loci, suggesting the observed genotype–phenotype association for MD may be mediated through gene expression. Further, a recent transcriptomic study[66] evaluating differentially expressed pathways in breast tissue samples from obese vs normal-weight adolescents, identified inflammation-related genes as among the most highly activated upstream regulators in the obese breast tissue samples.

Our study thoroughly explores the links between adiposity, puberty timing, and mammographic density, and breast cancer. The major finding of this study is that mammographic density, specifically dense area, potentially accounts for 56% of the protective effect of childhood adiposity on breast cancer. Understanding this mediating pathway is crucial since simply advocating for weight gain in childhood is clearly not a desirable goal. This finding is exciting because showing that adult MD is modifiable during the pubertal growth period means there could be opportunities to intervene during adolescence to reduce lifetime MD and associated breast cancer risk[18]. An important point to raise is that identifying a mediator of a causal relationship is just an initial step in dissecting it. MD is a relatively high-level concept, while the biological mechanistic pathways implicated in the overall process are still unknown. Therefore, an iterative approach to mediation studies may be appropriate, i.e., the next steps would be to focus on understating the causal pathway between MD and breast cancer and also between adiposity and MD. Further understanding of the underlying mechanism and biological pathways is required to explore potential avenues for intervention. In the study, we also showed that the density-increasing effect of later menarche may be due to lower adiposity in adolescence, which is associated with later puberty rather than an effect of age at menarche directly. The mechanisms linking childhood body size and age at menarche to breast cancer risk could, therefore, be entirely different and acting in opposing directions. Lastly, we

found that genetic instruments for MD are heterogeneous and pleiotropic, and there are likely several pathways underlying the role of mammographic density in influencing breast cancer risk. As MD GWAS sample sizes increase, this relationship can be further investigated, enhancing our understanding of the genetic basis of MD and its role in the aetiology of breast cancer.

## Methods

### Data sources

The mammographic density GWAS data used in this study is a meta-analysis of two studies (Hologic study, $N = 20,311$ and GE study, $N = 3881$; in total $N = 24,192$) of non-Hispanic white women aged between 40 and 74 years (~80% post-menopausal) from a larger population-based study, RPGEH (Research Program on Genes, Environment and Health), administered by Kaiser Permanente Northern California (KPNC) Division of Research[67,68]. Written informed consent was obtained from all participants. Institutional Review Board approvals for this study were obtained from KPNC, Stanford University, and the Icahn School of Medicine at Mount Sinai. The cohort details and study design are available in the original publication of mammographic density GWAS data[32]. Genotypes were re-imputed with an expanded reference panel, including the Haplotype Reference Consortium in addition to the 1000 Genomes Project Phase III data, to improve accuracy for less common variants. The GWAS analyses were adjusted for age at mammogram, BMI, genotype reagent kit, and the first ten principal components of ancestry[32]. Three mammographic density phenotypes were analysed: dense area (DA), non-dense area (NDA), and percent density (PD). The original MD GWAS published by Sieh et al. in 2020[32] was adjusted for BMI. For this study, the GWAS was rerun without this adjustment ("unadjusted GWAS") on a slightly smaller subset of 24,158 women from the original cohort.

Childhood body size, adult body size, and age at menarche data used in this study were obtained from UK Biobank[69]. All individual participant data used from the UK Biobank study had ethical approval from the Research Ethics Committee (REC; approval number: 11/NW/0382) and informed consent from all enrolled participants. UK Biobank is a population-based health research resource consisting of approximately 500,000 people, aged between 40 and 69 years, who were recruited between 2006 and 2010 from across the UK. The study design, participants and quality control (QC) methods have been described in detail previously[69]. The GWAS of childhood body size and adult body size used in this study were performed by Richardson et al.[4] on UK Biobank data ($N = 246,511$; female-only data). Childhood body size is a categorical trait describing body size at age 10, with three categories ('thinner than average', 'about average', 'plumper than average'), from a questionnaire completed by adult participants of UK Biobank. Adult body size measure was converted from continuous adult BMI in UK biobank into three groups based on the proportions of childhood body size data to ensure that the GWAS results of both measures are comparable[4]. The genetic scores for childhood and adult body size were independently validated in three separate cohorts (the HUNT study (Norway)[70], Young Finns Study[71], and ALSPAC (UK)[4]), which confirmed that the genetic instruments extracted by Richardson et al.[4] can reliably separate childhood and adult body size as distinct exposures, in addition to being robust to differential measurement error in simulations performed in the original study. Age at menarche GWAS summary data ($N = 143,819$) was accessed through OpenGWAS[72] (gwas.mrcieu.ac.uk) under ID *ukb-b-3768*.

The breast cancer data used in the study is from the Breast Cancer Association Consortium (BCAC) cohort of 2017 ($N = 228,951$; overall sample and ER+/ER- samples, assessed from OpenGWAS under IDs: *ieu-a-1126, ieu-a-1127, ieu-a-1128*)[34] and the latest release of BCAC in 2020 ($N = 247,173$; overall sample and five molecular subtypes: Luminal A, Luminal B1 (HER+), Luminal B2 (HER-), HER2-enriched, and triple-negative breast cancer)[35] (details in Supplementary Table 1). The cohort

design and genotyping protocol details are described elsewhere (bcac.ccge.medschl.cam.ac.uk/bcac-groups/study-groups/, bcac.ccge.medschl.cam.ac.uk/bcacdata/). The study groups in the BCAC cohort do not include UK Biobank or MD GWAS cohorts. The overall sample results presented throughout the paper are for BCAC 2017 data. The results for BCAC 2020 overall sample are available in all relevant Supplementary tables, and are not shown here due to their similarity.

### Mendelian randomization

Mendelian randomization (MR) is an application of instrumental variable analysis where genetic variants are used as instruments to estimate the causal relationship between a modifiable health exposure and a disease outcome[8,9]. There are three core assumptions that genetic variants need to satisfy to qualify as valid instruments for the causal inference: (1) variants have to be reliably associated with exposure of interest, (2) there cannot be any confounders of the instrument and the outcome, and (3) variants cannot be independently associated with the outcome, via pathway other than through the exposure (i.e. horizontal pleiotropy)[73].

The analyses in this work were performed using the two-sample (univariable) MR approach, which relies on using GWAS summary statistics of two non-overlapping samples for exposure and outcome[74]. Two-sample MR analyses were performed using the inverse-variance weighted (IVW) method[75]. Alongside IVW, other complementary MR methods were applied to assess the robustness of the causal estimates and to overcome any potential violations of the MR assumptions (e.g. horizontal pleiotropy) (see **Sensitivity analysis** for further details).

We used the two-step MR framework to assess whether an intermediate trait acts as a causal mediator between the exposure and the outcome of interest[76,77]. Multivariable Mendelian randomization (MVMR) was used to estimate the independent direct effects of two traits together on the outcome[78,79]. The genetic variants included in MVMR analysis have to be reliably associated with one or both exposures but not completely overlap (i.e. no perfect collinearity), and have to satisfy the MVMR-extended second and third assumptions of the standard MR analysis[45]. Diagnostic methods and sensitivity tests for the robustness of MVMR estimates[45,80] are described under **Sensitivity analysis**.

All analyses were conducted using R (v4.2.1). Univariable MR analyses and sensitivity tests were performed using the *TwoSampleMR* R package (v0.5.6)[81], which was also used for accessing GWAS summary data deposited in OpenGWAS[72] (gwas.mrcieu.ac.uk). Multivariable MR was carried out using the *MVMR* R package (v0.2)[78].

For all exposure traits, the instruments were extracted by selecting SNPs with $p$-value under the $5 \times 10^{-8}$ threshold and clumping the resulting set of variants with r2 = 0.001 using the default LD (linkage disequilibrium) reference panel in *TwoSampleMR* (1000 Genomes Project, European data only). When extracting instruments from the outcome (breast cancer) GWAS summary statistics, unavailable SNPs were substituted by proxies with a minimum LD r2 = 0.8. The rest of the settings were kept to defaults as per the package version number. The number of instruments used in the analysis for all exposures: childhood body size ($n = 115$), adult body size ($n = 173$), age at menarche ($n = 190$), dense area ($n = 21$), non-dense area ($n = 8$), percent density ($n = 11$).

### Sensitivity analysis

In addition to the standard MR analysis (IVW), we used MR-Egger[42] and weighted median[43] MR methods to evaluate the validity of the analysed genetic instruments and to overcome and accommodate potential violations of the core MR assumptions. These complementary methods help to support the causal effects found with IVW, as a single method cannot account for all biological and statistical properties that may impact MR estimates. Also a variety of specialised tests were applied, as recommended in ref. 81.

To assess overall horizontal pleiotropy (violation of assumption 3), the intercept in the MR-Egger regression[42] was evaluated, and the

heterogeneity among the genetic variants was quantified using Cochran's Q-statistic[44]. The intercept term in MR-Egger regression is a useful indication of whether directional horizontal pleiotropy is driving the results of an MR analysis, under the assumption that any pleiotropic effects are uncorrelated with the magnitude of the SNP exposure association. When the Egger intercept is close to zero (e.g. < 0.002) and the P-value is large, this can be interpreted as no evidence of a substantial directional (horizontal) pleiotropic effect.

When the Q-statistic for heterogeneity (difference in individual ratio estimates) is high and the corresponding p-value is small, this suggests evidence for heterogeneity and possibly horizontal pleiotropy. A high Q-statistic can also be used as an indicator of one or more variant outliers in the analysis, which may also violate the MR assumptions. In univariable MR, heterogeneity may be indicative of horizontal pleiotropy that does not act through one of the exposures. In MVMR, heterogeneity is quantified by $Q_A$-statistic (also a further modification of Cochran's Q), and small $Q_A$ indicates a lack of heterogeneity in the per-SNP effects[45].

We derived F-statistics in both univariable and MVMR to evaluate the instrument strength[45,82], with $F > 10$ indicating sufficient strength for minimal weak instrument bias in the analysis. We also evaluated the possibility of reverse causation via Steiger filtering and assessed whether each instrument explains more variance ($R^2$) in the exposure rather than in the outcome[83].

### Additional sensitivity and outlier analyses

To explore the excessive heterogeneity and potential pleiotropy identified in the effect of MD on breast cancer, we explored the genetic instruments using several outlier detection methods.

First, we applied MR-PRESSO[36], a method that detects overall pleiotropic bias through outlier detection by assessing each genetic variant's contribution to the overall heterogeneity. This method discards influential outliers from the IVW method and uses a distortion test to evaluate the significance of the distortion between the causal estimate before and after the removal of the outlier variants, providing an outlier-corrected pleiotropy-robust causal estimate as a result. The analysis was run using the *MR-PRESSO* R package (v1.0), using the default parameters.

We also used the approach implemented in *Radial-MR*[38] (R package v1.0) to identify outliers with the most weight in the MR analysis and the largest contribution to Cochran's Q statistic for heterogeneity. The analysis was conducted with a p-value threshold (*alpha* parameter) set to Bonferroni corrected for the number of SNPs tested in the analysis ($p < 0.05$/number of instruments in the exposure) and using modified second-order weights (*weight* parameter).

Finally, to investigate the presence of clustered heterogeneity and assess the possibility of there being several distinct causal mechanisms by which MD may influence breast cancer risk, we performed clustered Mendelian randomization using *MR-Clust*[37] (R package v0.1.0). MR-Clust is a heterogeneity-based clustering algorithm that extends the typical MR assumption that a risk factor can influence an outcome via a single causal mechanism[84] to a framework that allows one or more mechanisms to be detected. The heterogeneity and outliers in the main MR result may indicate that different genetic variants influence the risk factor in distinct ways, e.g., via distinct biological mechanisms.

MR-Clust assigns variants to $K$ clusters, where all variants have similar causal ratio estimates, a "null" cluster (variants with a null effect), and a "junk" cluster (non-null variants that do not fit into any of the K clusters). In our analysis, the clusters were formed of variants that had a great conditional probability of assignment (score > 0.9), keeping the results conservative. Due to the limited number of instruments in MD exposure, we kept all clusters regardless of their size (visualised using the MR-Clust package built-in scatter plot).

The outliers identified by MR-PRESSO and Radial-MR analyses, as well as clusters of SNPs detected by MR-Clust, were displayed using single-SNP forest plots to explore individual SNPs heterogeneity. The single-SNP forest plots show the effect of the exposure on the outcome for each SNP separately (i.e. Wald ratio). The plots also included the IVW MR estimate with the identified outliers excluded, and the individual estimates for identified clusters.

### PheWAS

To further examine the genetic instruments of the MD phenotypes and better understand the potential sources of effect heterogeneity, we performed a phenome-wide association study (PheWAS) analysis[39]. We used PhenoScanner V2 (*phenoscanner* R package v1.0)[85,86] and Open-GWAS database (gwas.mrcieu.ac.uk/phewas/, accessed via *ieugwasr* R package v0.1.5)[72] to query publicly available GWAS data for associations with the SNPs from the MD phenotypes. The query was restricted to European ancestry datasets, retrieving SNP-trait associations of p-value $< 5 \times 10^{-8}$ and adjusting for FDR.

We presented PheWAS results for each MD SNP grouped by clusters determined by the MR-Clust algorithm. This helped us to review the association differences between clusters of SNPs with the traits identified in GWAS databases, which might explain some of the observed heterogeneity in the MR results.

### Gene and pathway exploration

To explore the functional relevance of the identified clusters of MD instruments, we mapped instrument SNPs of each MD phenotype to genes and identified the pathways they are involved in. For gene mapping, we used the SNP2Gene function on the FUMA (the Functional Mapping and Annotation of GWAS) platform[87], applying positional mapping (500 kb) and eQTL-based mapping (including only GTEx v8 breast or adipose tissue datasets). All default settings were applied, including the eQTL p-value threshold for significant snp-gene pairs (FDR < 0.05), to find genes whose expression was associated with the locus of instrument SNPs. The pathways were extracted using the *enrichR* R package (v3.1)[88] (including pathway definitions from Reactome, KEGG, GO terms, and WikiPathway databases). We also used the *ReactomeContentService4R* R package (v1.4.0)[89] to obtain more recent Reactome data (data accessible from enrichR is pre-2016). The pathway data was collected for a broader context only, and no formal gene-set overrepresentation analysis was performed.

### Mediation analysis

Mediation analysis is used to quantify the effects of an exposure on an outcome, which act directly or indirectly via an intermediate variable (i.e., mediator)[90]. Identifying mediators of the relationship between the exposure and the outcome enables intervention on those mediators to mitigate or strengthen the effects of the exposure[91].

The total effect of exposure on outcome includes both a direct effect and any indirect effects via one or more mediators. The total effect is captured by a standard univariable MR analysis. To decompose direct and indirect effects, we used the results from two-step MR and MVMR in two mediation analysis methods: Difference method and Product method.

For the Difference method, to estimate the indirect effect, we subtracted the direct effect of exposure on the outcome from MVMR (in analysis with the mediator) from the total effect of exposure on the outcome (univariable MR)[55]. In Product method (also known as 'product of coefficients'), the results from two steps of two-step MR analysis (i.e., the effect of exposure on the mediator and the effect of the mediator on the outcome) are multiplied to get the indirect effect[76,92]. Here, we used the direct effect of the mediator on the outcome from MVMR as the second term in the calculation[90]. To estimate the standard error (SE) and later confidence intervals (CIs) of the indirect effect, we used 'Propagation of errors' approach for the Difference method estimate (as outlined in ref. 55) and Delta method (also known as Sobel test[93]) for the Product method estimate. Further details on

performing mediation analysis are available in the Supplementary materials of our previous work[10]). The mediation analysis calculations are presented in Supplementary Note 2.

## Reporting summary

Further information on research design is available in the Nature Portfolio Reporting Summary linked to this article.

## Data availability

The GWAS data for BCAC 2017 breast cancer (IDs: *ieu-a-1126, ieu-a-1127, ieu-a-1128*), age at menarche (*ukb-b-3768*), and childhood body size (*ieu-b-510*), can be accessed from OpenGWAS[72] (https://gwas.mrcieu.ac.uk). The BCAC 2020 molecular subtype data was published by Zhang et al.[35], and is available at https://bcac.ccge.medschl.cam.ac.uk/bcacdata/. Childhood and adult body size GWAS data was published by Richardson et al.[4]. This study uses data from a GWAS of mammographic density (published by Sieh et al.[32]) The RPGEH genotype data are available upon application to the KP Research Bank (https://researchbank.kaiserpermanente.org/). Additional relevant information (e.g. genetic association data) is available from Sieh et al.[32] or the authors upon request. Source data for all figures are provided with this paper. Source data are provided with this paper.

## Code availability

All analyses in this study are available at: https://github.com/mvab/mammographic_density_mr, https://doi.org/10.5281/zenodo.10724802.

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

## Acknowledgements

M.V. is supported by the University of Bristol Alumni Fund (Professor Sir Eric Thomas Scholarship). B.L.L. is supported by the University of Bristol Vice-Chancellor's fellowship, Academy of Medical Sciences, Elizabeth Blackwell Institute for Health Research (University of Bristol) and the Wellcome Trust Institutional Strategic Support Fund (ISSF3 (204813/Z/16/Z) and AMS (SBF003/1170)). M.V., G.D.S., E.S., T.G.R., R.C.R. work in the UK Medical Research Council Integrative Epidemiology Unit at the University of Bristol supported by Medical Research Council (MC_UU_00032/01, MC_UU_00032/03, MC_UU_00032/04). This work is also supported by a Cancer Research UK programme grant (the Integrative Cancer Epidemiology Programme) (C18281/A29019). W.S., L.A.H., J.H.R., S.E.A. are supported by the U.S. National Institutes of Health (R01CA237541, R01CA264987, R01CA166827). We would also like to acknowledge Tom Gaunt and Tim Robinson for thoughtful project discussions.

## Author contributions

M.V., B.L.L., R.C.R., G.D.S., T.G.R., E.S. conceived and designed the study. M.V. performed the analyses, interpreted the results, and wrote the initial draft of the manuscript as well as subsequent drafts with critical input on results interpretation and manuscript revisions from E.S., B.L.L., T.G.R., G.D.S., R.C.R., W.S., L.A.H., J.H.R., S.E.A. Access to mammographic density data was provided by W.S., L.A.H., J.H.R., S.E.A.; J.H.R. performed genome-wide association studies of mammographic density phenotypes unadjusted for BMI.

## Competing interests

T.G.R. is employed by GSK outside of this work, for unrelated research. All other authors declare no competing interests.
