## [Peer Review File · Nature Communications]

Mammographic density mediates the protective effect of early-life body size on breast cancer riskReviewer #1, expertise in risk analysis epidemiology and obesity (Remarks to the Author):

This is an interesting study, using advanced MR methodology, to examine the potential mediating role of imaging-based density measures of the association between child obesity and breast cancer risk. The research group is well known, the research was well conducted, and the study reads well. I have a few comments/questions that need to be addressed during revisions.

1) The childhood adiposity GWAS was performed in men and women combined, but the breast cancer GWAS and density GWAS are done in women only. Given the differences between boys and girls during childhood with respect to their adiposity, it is safe to assume there is no difference and you can use these instruments as done by the authors? Perhaps run some sex-SNP interaction analyses on childhood adiposity?

2) The authors performed first a mediation analysis and subsequently argue that there is substantial heterogeneity that could be explained by different mechanisms. As the authors mention nothing about this in the abstract on this, I wonder whether this mediating effect of 56% is still valid. Please comment on this. Perhaps add mediation analyses using the information from the clusters?

3) Do the authors have access to the DA/NDA/PD data; perhaps lowering the threshold (but still keep strong instruments) is of added value here?

4) How would age of menopause fit in this whole process?

Reviewer #2, expertise in breast cancer GWAS (Remarks to the Author):

Vabistsevits and colleagues have conducted comprehensive analyses to investigate the mediating role of mammographic density (MD) in the protective effect of early-life adiposity on breast cancer risk using the Mendelian randomization (MR) framework. Employing the MR approach is a well-accepted method to discern potential causal effects of exposure on disease/trait outcomes by leveraging robust genetic predictors as instrumental variables.

In their study, the authors analyzed Genome-Wide Association Studies (GWAS) datasets for childhood body size, adult body size, age at menarche, MD (including Dense area [DA], non-dense area [NDA], and percent density [PD]), and breast cancer, encompassing various subtypes. Initially, they applied the Inverse Variance Weighted (IVW) MR analysis to assess the impact of MD on breast cancer risk. It was noted that the precision of the effect estimation might be affected by heterogeneity among the genetic instruments for the MD phenotypes. To address this, the authors subsequently employed MR-PRESSO and Radial-MR approaches to identify and correct for variant outliers. This outlier-corrected IVW MR analysis revealed statistically significant effects of MD phenotypes on breast cancer risk. Furthermore, their mediation analysis demonstrated that higher childhood adiposity is associated with a reduced risk of breast cancer, which is mediated through reductions in mammographic DA.

While this study is intriguing and holds significant relevance in the field of genetic epidemiology, certain aspects of the analysis description warrant further clarification to substantiate their analytical strategy. Specific comments are provided below:

The authors should provide information on Linkage Disequilibrium and GWAS P-values for the selected SNPs to establish the independence and strength of the genetic instruments employed in the MR analysis.

It is imperative for the authors to present the results of the outlier-corrected IVW MR analysis for the effects of MD phenotypes on distinct breast cancer subtypes. This will be important to evaluate the results from the outlier-corrected IVW MR analysis.

The exploration of genes and pathways pertinent to the MD phenotype using the selected genetic variants as instruments is intriguing. However, it remains unclear whether these variants are indeed GWAS-identified variants, and the process for identifying target genes based on Expression Quantitative Trait Loci (e.g., p-values) requires clarification. Considering additional functional

genomic analyses, such as gene-enhancer link analysis, could enhance the comprehensiveness of the study.

Responses to reviewers

Please find below our detailed responses to all comments and suggestions marked in blue. *Italicised* text indicates text quoted from the manuscript. The line numbers mentioned in the responses refer to the revised text. All changes in the revised text are tracked.

Reviewer #1, expertise in risk analysis epidemiology and obesity (Remarks to the Author):

This is an interesting study, using advanced MR methodology, to examine the potential mediating role of imaging-based density measures of the association between child obesity and breast cancer risk. The research group is well known, the research was well conducted, and the study reads well. I have a few comments/questions that need to be addressed during revisions.

⇒ Thank you for the positive appraisal of the manuscript and for raising several valid queries. We have provided detailed clarifications for them below.

1) The childhood adiposity GWAS was performed in men and women combined, but the breast cancer GWAS and density GWS are done in women only. Given the differences between boys and girls during childhood with respect to their adiposity, it is safe to assume there is no difference and you can use these instruments as done by the authors? Perhaps run some sex-SNP interaction analyses on childhood adiposity?

The childhood adiposity GWAS was performed using female-only data, including for instrument extraction (line 686 in Methods):

“The GWAS of childhood body size and adult body size used in this study were performed by Richardson et al [1] on UK Biobank data (N= 246,511; female-only data).”

We apologise that this was unclear from the text, so now we also mention this in Table 1 legend that describes all conducted analyses.

Your concern regarding using mixed-sex same in this analysis is valid, but as our breast cancer-focused study has originally been designed to use female-only data, we have not explored whether there would be differences in the effect by sex.

2) The authors performed first a mediation analysis and subsequently argue that there is substantial heterogeneity that could be explained by different mechanisms. As the authors mention nothing about this in the abstract on this, I wonder whether this mediating effect of 56% is still valid. Please comment on this. Perhaps add mediation analyses using the information from the clusters?

Thank you for making this point. The heterogeneity in the individual SNP effects does affect the confidence intervals around the estimated mediated effect (56% [95% CIs 32%-79%]). However, if there were less heterogeneity, this result would still hold, but the estimate would be more precise.

To demonstrate this, we performed mediation calculation using the outlier-corrected effect estimates (i.e. using SNPs equivalent to cluster_2 in Figure 4d), which are less heterogeneous. We found that the proportion mediated via DA using this heterogeneity-corrected subset of SNPs is 45% [24% - 66%]. The new proportion-mediated estimate falls within the CIs range of the main analysis, and is more precise. This suggests that the originally reported estimate of 56% [32%-79%] is valid, and the

wide CIs highlight the heterogeneity of the results. The calculations are provided in Supplementary Note 2.2.

We mention this additional analysis in *Results/Mediation analysis* section of the manuscript (lines 475-478, below) and MD SNPs heterogeneity is highlighted in the abstract.

“As DA instruments were found to be heterogeneous, we also performed mediation analysis using outlier-corrected estimates to assess the validity of the proportion-mediated effect. The heterogeneity-corrected estimate falls into the CIs of the main results and is more precise (see Supplementary Note 2.2).”

3) Do the authors have access to the DA/NDA/PD data; perhaps lowering the threshold (but still keep strong instruments) is of added value here?

Thank you for the suggestion. We made the decision to use the genome-wide significance threshold for selecting SNPs to ensure that they are robustly associated with MD phenotypes and to limit the risks of introducing weak-instrument bias and horizontal pleiotropy. Lowering the threshold would also likely add to already substantial heterogeneity. The currently selected instruments for MD phenotypes have good instrument strength (F-stat >40).

Additionally, for this study, we had access to a specific list of SNPs, which included genome-wide significant SNPs in DA/NDA/PD and the SNPs that were instruments in the other traits required for MVMR analyses.

4) How would age of menopause fit in this whole process?

We appreciate this comment, as it opens up a few possible ideas for discussion.

Age at menopause may have a role in the chain of biological processes/traits investigated in this work. However, in our previous work [2], we showed that age at menopause is unlikely to be a mediator between childhood adiposity and breast cancer, so we did not consider including it in this study. Investigating age at menopause may be relevant to understanding the relationship between mammographic breast density and breast cancer, as it is likely that shared hormonal and metabolic processes are involved. Including this analysis here would deviate too much from the main focus of this paper, but it is an interesting question for future work.

Menopause status is also important to consider when thinking about MD and breast cancer data stratification. In the cohort where MD was measured (N=24K), approximately 80% of women are post-menopausal [3]. In the original GWAS publication, analyses stratified by menopausal status showed that MD SNP effects were similar in pre-menopausal and post-menopausal women [3]. In future studies, when larger sample sizes are available, it might be interesting to look into this further, i.e. evaluate a menopause status-stratified effect of MD on breast cancer risk. As breast density naturally declines with age [4], and studies have shown that the magnitude of the association between density and breast cancer risk is stronger in pre-menopausal women than in post-menopausal women [5], we might be able to detect similar differences using MR.

The final point to consider is breast cancer stratification by menopause status. The BCAC breast cancer summary data is not available by menopause status, so testing the effect of MD directly on pre- and post-menopausal breast cancer is not possible. The ER-/ER+ subtypes might crudely proxy for pre/post-menopausal breast cancer, respectively, as younger women have ER-breast cancer more commonly [6]. However, as the MD effect is measured in a predominantly post-menopausal sample, evaluating its effect on proxy-samples may be over-interpreting the results. Moreover, the MD effect on all available breast cancer subtypes looks generally consistent (Figure 3). Any observed differences may not be significant, as all estimates have

substantial heterogeneity, and the slight differences could be related to results imprecision due to a lower percentage of cases in ER- subtypes data (Supplementary Table S1).

To address the importance of considering menopause status in this work, we made the following changes to the manuscript:

- Made it clear in MD GWAS data description that MD was measured in pre-dominantly post-menopausal women: *Methods/Data sources* (line 668)
- Mentioned in the Discussion that a larger sample size that includes pre-menopausal women might let us do a menopause status-stratified analysis in future studies (lines 617-619).

Reviewer #2, expertise in breast cancer GWAS (Remarks to the Author):

Vabistsevits and colleagues have conducted comprehensive analyses to investigate the mediating role of mammographic density (MD) in the protective effect of early-life adiposity on breast cancer risk using the Mendelian randomization (MR) framework. Employing the MR approach is a well-accepted method to discern potential causal effects of exposure on disease/trait outcomes by leveraging robust genetic predictors as instrumental variables.

In their study, the authors analysed Genome-Wide Association Studies (GWAS) datasets for childhood body size, adult body size, age at menarche, MD (including Dense area [DA], non-dense area [NDA], and percent density [PD]), and breast cancer, encompassing various subtypes. Initially, they applied the Inverse Variance Weighted (IVW) MR analysis to assess the impact of MD on breast cancer risk. It was noted that the precision of the effect estimation might be affected by heterogeneity among the genetic instruments for the MD phenotypes. To address this, the authors subsequently employed MR-PRESSO and Radial-MR approaches to identify and correct for variant outliers. This outlier-corrected IVW MR analysis revealed statistically significant effects of MD phenotypes on breast cancer risk. Furthermore, their mediation analysis demonstrated that higher childhood adiposity is associated with a reduced risk of breast cancer, which is mediated through reductions in mammographic DA.

While this study is intriguing and holds significant relevance in the field of genetic epidemiology, certain aspects of the analysis description warrant further clarification to substantiate their analytical strategy. Specific comments are provided below:

- ⇒ Thank you for the positive appraisal of the manuscript, we appreciate the valuable feedback. We provide the responses to your queries below and refer to incorporated changes from your suggestions by line numbers.

1) The authors should provide information on Linkage Disequilibrium and GWAS P-values for the selected SNPs to establish the independence and strength of the genetic instruments employed in the MR analysis.

We provide information on the instrument extraction, including p-value threshold and the LD panel used for clumping, in a paragraph starting on line 744 in Methods:

“For all exposure traits, the instruments were extracted by selecting SNPs with p-value under the 5×10^{-8} threshold and clumping the resulting set of variants with $r^2 = 0.001$ using the default LD (linkage disequilibrium) reference panel in TwoSampleMR (1000 Genomes Project, European data only). When extracting instruments from the outcome (breast cancer) GWAS summary statistics, unavailable SNPs were substituted by proxies with a minimum LD $r^2 = 0.8$. The rest of the settings were kept to defaults as per the package version number.”

The used instruments are independent and strong; in MR analysis we also calculated F-statistics for MD phenotypes, which were all > 10 (Supplementary Table S12).

2) It is imperative for the authors to present the results of the outlier-corrected IVW MR analysis for the effects of MD phenotypes on distinct breast cancer subtypes. This will be important to evaluate the results from the outlier-corrected IVW MR analysis.

Thank you for bringing up this point; we agree that reviewing the consistency of outliers in the MD effect on different breast cancer subtypes would add further confidence to our results. We identified outliers and obtained outlier-corrected IVW estimates for mammographic DA effect on all breast cancer subtypes using Radial-MR. The single SNP forest plots for each subtype with outliers highlighted are included below (equivalent to Figure 4b in the manuscript). The SNPs in plots are ordered according to the SNP order in the overall breast cancer results (i.e. by SNPs effect size in plot panel a; same as Figure 4b) to assist with outlier comparison.

First, we see that individual SNP estimates are more precise in the analyses where breast cancer samples are larger (i.e. overall samples, ER+/ER-, and Luminal A). For subtypes with smaller sample sizes (and lower % of cases, Supplementary Table S1), the individual SNP estimates are quite imprecise: HER2, Luminal B1/2, TNBC. Moreover, for larger sample sizes, the same set of SNPs are consistently found to be outliers. For other subtypes, due to the overall imprecision of estimates, fewer SNPs are flagged as outliers, and they are less consistent. Still, all of the identified outliers in subtypes match the ones found in the main analysis. The overall outlier-corrected IVW estimates are consistent among the subtypes.

In the manuscript, we refer to these additional results on line 299. This analysis is added as Supplementary Note 3, and the corresponding data is available in Supplementary Table S17.

“In this investigation, we focused on the overall breast cancer sample outcome as the main analysis, but additional results for breast cancer subtypes are available in Supplementary Note 3.”

a Single SNP forest plot with outliers

b Single SNP forest plot with outliers

c Single SNP forest plot with outliers

d Single SNP forest plot with outliers

e Single SNP forest plot with outliers

f Single SNP forest plot with outliers

g Single SNP forest plot with outliers

h Single SNP forest plot with outliers

i Single SNP forest plot with outliers

3) The exploration of genes and pathways pertinent to the MD phenotype using the selected genetic variants as instruments is intriguing. However, it remains unclear whether these variants are indeed GWAS-identified variants, and the process for identifying target genes based on Expression Quantitative Trait Loci (e.g., p-values) requires clarification. Considering additional functional genomic analyses, such as gene-enhancer link analysis, could enhance the comprehensiveness of the study.

Thank you for these points. To address each one:

- a) The variants we used in the gene/pathway exploration are indeed GWAS-identified instrument SNPs we used in MR analysis. We apologise that that was not clearly stated. We updated the text on line 389 to make it more obvious:

“To gain some biological context for the identified outlier SNPs and distinct clusters among the MD instruments, we mapped the variants used in MR analyses to genes.”

- b) We apologise that insufficient detail was provided regarding identifying target genes. We've updated the section (line 847) to include more detail:

“For gene mapping, we used the SNP2Gene function on the FUMA (the Functional Mapping and Annotation of GWAS) platform, applying positional mapping (500kb) and eQTL-based mapping (including only GTEx v8 breast or svalue threshold for significant snp-gene pairs (FDR < 0.05), to find genes whose expression was associated with the locus of instrument SNPs.”

- c) Thank you for recommending the additional type of functional analysis.

The original MD GWAS paper [3], where the genetic instruments for our MR analyses were taken from, provides an in-depth functional follow-up of the identified MD-associated SNPs. In addition to positional and gene expression-based mapping, the authors also assessed the lead SNPs location in the promoter or enhancer regions in mammary epithelial cells and fibroblasts using data from the ENCODE and Roadmap Epigenomic consortia (see details in section 'Regulatory function of MD SNPs' and 'Functional analyses' in Sieh *et al* <https://doi.org/10.1038/s41467-020-18883-x>).

In this study, we performed the necessary gene mapping that helped us to put the SNP clustering results into some context. We believe, however, that further functional follow-up adds little to that already described in the original MD GWAS paper and risks diluting the analysis and overarching message of this manuscript. For readers with a more specialised interest in the functional role of MD variants, reviewing the results of the original GWAS publication [3] may be more appropriate.

References

- [1] T. G. Richardson, E. Sanderson, B. Elsworth, K. Tilling, and G. Davey Smith, "Use of genetic variation to separate the effects of early and later life adiposity on disease risk: Mendelian randomisation study," *The BMJ*, vol. 369, 2020, doi: 10.1136/bmj.m1203.
- [2] M. Vabistsevits, G. Davey Smith, E. Sanderson, T. G. Richardson, B. Lloyd-Lewis, and R. C. Richmond, "Deciphering how early life adiposity influences breast cancer risk using Mendelian randomization," *Commun Biol*, vol. 5, no. 1, 2022, doi: 10.1038/s42003-022-03272-5.
- [3] W. Sieh *et al.*, "Identification of 31 loci for mammographic density phenotypes and their associations with breast cancer risk," *Nat Commun*, vol. 11, no. 1, Dec. 2020, doi: 10.1038/s41467-020-18883-x.
- [4] A. Burton *et al.*, "Mammographic density and ageing: A collaborative pooled analysis of cross-sectional data from 22 countries worldwide," *PLoS Med*, vol. 14, no. 6, p. e1002335, Jun. 2017, doi: 10.1371/JOURNAL.PMED.1002335.
- [5] L. Yaghjian, G. A. Colditz, B. Rosner, and R. M. Tamimi, "Mammographic breast density and breast cancer risk by menopausal status, postmenopausal hormone use and a family history of breast cancer," *Cancer Causes Control*, vol. 23, no. 5, pp. 785–790, May 2012, doi: 10.1007/S10552-012-9936-7.
- [6] L. Chollet-Hinton *et al.*, "Breast cancer biologic and etiologic heterogeneity by young age and menopausal status in the Carolina Breast Cancer Study: A case-control study," *Breast Cancer Research*, vol. 18, no. 1, pp. 1–10, 2016, doi: 10.1186/s13058-016-0736-y.

Reviewer #1 (Remarks on code availability):

Yes.

Reviewer #2 (Remarks to the Author):

The authors have satisfactorily addressed all of my comments and concerns.